# Preeclampsia and the Antiphospholipid Syndrome

**DOI:** 10.3390/biomedicines11082298

**Published:** 2023-08-18

**Authors:** Karoline Mayer-Pickel, Manurishi Nanda, Maja Gajic, Mila Cervar-Zivkovic

**Affiliations:** Department of Obstetrics, Medical University Graz, 8036 Graz, Austria; manurishi.nanda@medunigraz.at (M.N.); maja.gajic@medunigraz.at (M.G.); mila.cervarzivkovic@medunigraz.at (M.C.-Z.)

**Keywords:** obstetric antiphospholipid syndrome, preeclampsia, prematurity, alternative therapy, vitamin D, curcumin, hydroxychloroquine

## Abstract

Antiphospholipid syndrome (APS) is characterized by venous or arterial thrombosis and/or adverse pregnancy outcome in the presence of persistent laboratory evidence of antiphospholipid antibodies (aPLs). Preeclampsia complicates about 10–17% of pregnancies with APS. However, only early onset preeclampsia (<34 weeks of gestation) belongs to the clinical criteria of APS. The similarities in the pathophysiology of early onset preeclampsia and APS emphasize an association of these two syndromes. Overall, both are the result of a defective trophoblast invasion and decidual transformation at early gestation. Women with APS are at increased risk for prematurity; the reasons are mostly iatrogenic due to placental dysfunction, such as preeclampsia or FGR. Interestingly, women with APS have also an increased risk for preterm delivery, even in the absence of FGR and preeclampsia, and therefore it is not indicated but spontaneous. The basic treatment of APS in pregnancy is low-dose aspirin and low-molecular-weight heparin. Nevertheless, up to 20–30% of women develop complications at early and late gestation, despite basic treatment. Several additional treatment options have been proposed, with hydroxychloroquine (HCQ) being one of the most efficient. Additionally, nutritional interventions, such as intake of vitamin D, have shown promising beneficial effects. Curcumin, due to its antioxidant and anti-inflammatory properties, might be considered as an additional intervention as well.

## 1. Introduction

Antiphospholipid syndrome (APS) is a systemic autoimmune disease characterized by the presence of antiphospholipid antibodies/aPLs (anticardiolipin antibodies/ACLA, lupus anticoagulant/LA, and anti-ß2-glycoprotein/anti-ß2-GPI). These antibodies are associated with arterial and/or venous thromboses and with various complications in early and late pregnancy [1,2].

Patients with solely obstetric complications but without any thrombotic events in the past are described as patients with obstetric antiphospholipid syndrome (OAPS) [3,4]. Clinical manifestations of OAPS are complications in early and late gestation, including recurrent early fetal loss before 10 weeks of gestation; late fetal loss at or beyond 10 weeks of gestation; or placental dysfunction in second and third trimester, such as fetal growth restriction, preeclampsia, or fetal death [1,2].

Recurrent (early) fetal loss is with a prevalence of up to 40% the main clinical manifestation of OAPS [5]; on the other hand, 20% of all women with recurrent fetal loss are tested positive for at least one antiphospholipid antibody (aPL) [6].

Late fetal loss and intrauterine fetal death (IUFD) are rather rare pregnancy complications in women with APS. Nevertheless, a significant association of IUFD and aPLs could be detected in up to 14% of cases [7,8,9].

Fetal growth restriction (FGR) is usually defined as an estimated fetal weight (EFW) or an abdominal circumference (AC) < 10th percentile for gestational age [10].

It is associated with an increased risk of perinatal morbidity and mortality, as well as impairment of long-term outcome [11]; additionally, the rate of preterm delivery is significant higher in fetuses with impaired fetal growth [12].

Preeclampsia is a pregnancy-specific multiorgan disorder, complicating 3–5% of all pregnancies [13]. Preeclampsia is diagnosed based on a new onset of hypertension and proteinuria or end organ damage after 20 weeks of gestation [13]. It presents itself in two different phenotypes, depending on the onset of symptoms as well as clinical manifestation [13]. Early onset of preeclampsia/EOP occurs before 34 weeks of gestation, and late onset preeclampsia/LOP after 34 weeks of gestation [13].

EOP is usually associated with fetal growth restriction and is considered the more “severe” form, whereas LOP usually is characterized by solely maternal manifestation and features a mild to moderate manifestation [14].

Prematurity is defined as a birth that occurs before 37 completed weeks of gestation [15]. It is associated with an increased neonatal morbidity and mortality, particularly among extremely preterm infants (<28 weeks) [15].

It is a known fact that women with APS have an increased risk of developing preeclampsia [16]; in these instances, preeclampsia is often severe and sometimes occurs even at the mid-trimester [16,17,18,19,20,21].

However, only preeclampsia, i.e., prematurity occurring before 34 weeks of gestation, is considered a manifestation of “classic” OAPS. When developing after 34 weeks of gestation, the Sydney criteria are not fulfilled, and the terms “aPL-related obstetric morbidity” (OMAPS) or non-criteria APS (NC-OAPS) should be used instead [3]. Other criteria of OMAPS are “only” two consecutive early fetal losses, at least two non-consecutive fetal losses, or premature rupture of fetal membranes [3].

The basic treatment of APS in pregnancy is low-dose aspirin (LDA) and low-molecular-weight heparin (LMWH). However, approximately 20–30% of women will suffer from pregnancy complications in spite of this treatment [22]. An interesting fact is that the treatment is more efficient for the prevention of preeclampsia or other forms of placental dysfunction in comparison to recurrent fetal loss [23]. Several additional treatment options have been proposed, with hydroxychloroquine (HCQ) being one of the most efficient [24,25,26,27].

Additionally, nutritional interventions, such as intake of vitamin D, have shown promising beneficial effects [28,29,30,31,32].

However, little is still known about the relationship of preeclampsia and APS. In particular, does prematurity occur without preeclampsia or other forms of placental dysfunction? Does late onset preeclampsia in women with APS (or aPL-positive women) even exist? And are there any other additional treatment options?

The answers to these questions are the focus of this review.

## 2. Discussion

Strictly speaking, preeclampsia is not an actual diagnostic criterion for APS, but prematurity before 34 weeks of gestation is [1]. Reasons for prematurity, i.e., preterm delivery, are mostly iatrogenic due to placental dysfunction, such as preeclampsia or FGR [1].

Preeclampsia is defined as a new onset of hypertension and proteinuria or the new onset of hypertension plus significant end-organ dysfunction with or without proteinuria in a previously normotensive patient, typically after 20 weeks of gestation or postpartum [33].

Preeclampsia presents itself in two different phenotypes, depending on onset of disease [14,34,35,36]. EOP (<34 weeks of gestation) is characterized by fetal (i.e., fetal growth restriction/FGR) and maternal manifestation, whereas in LOP (>34 weeks of gestation) maternal symptoms occur mostly alone [14]. Although EOP is often considered the more severe form and LOP is characterized by “only” mild to moderate manifestations, the importance of the consequences in both forms after delivery should not be underestimated.

### 2.1. Association of APS and Preeclampsia

The first relationship between aPLs, namely, lupus anticoagulant and complications in pregnancy, was described in 1980 by Firkin, B.G. et al. [37]. Three years later, a detailed clinical description of a new autoimmune syndrome—the antiphospholipid syndrome—was published for the first time by Graham Hughes [38]. An association of lupus anticoagulant and preeclampsia, i.e., FGR, was mentioned by Branch, D.W. in 1985 [39]. Due to a rather high rate of preeclampsia in women with aPLs, a first possible relationship of preeclampsia and aPLs have therefore been described by various authors [39,40,41,42,43].

#### 2.1.1. Pathophysiology of APS and Preeclampsia

The similarities in the pathophysiology of early onset preeclampsia and APS emphasize an association of these two syndromes. Overall, both are the result of a defective trophoblast invasion and decidual transformation at early gestation.

The “two-hit” hypothesis is considered the most relevant explanation of the development of clinical manifestations of APS: aPLs (“first hit”) cause a pro-coagulant phenotype by cell activation (such as platelets, monocytes, endothelial cells); activation of the anticoagulant and fibrinolytic system; and enhanced inflammation, especially the activation of the complement system. This pro-thrombotic state leads to thrombosis (“second hit”) in the presence of an additional factor or trigger [44]. However, the “two-hit” assumption does not fully explain the pathogenesis of the various obstetric manifestations in APS [45]. According to the literature, aPLs have a direct effect on trophoblast cells, leading i.a. to enhanced apoptosis, defective proliferation, and decreased angiogenesis, as well as a negative effect on transformation of spiral arteries and on maturation and differentiation of maternal decidual endometrial cells [44,46,47].

Preeclampsia is thought to be a syndrome, rather than a disease, with manifestation in various organs with systemic endothelial dysfunction. The exact mechanisms are still unclear, although several systemic processes have been described, and they play an important role: angiogenic imbalance, oxidative stress, and exaggerated systemic inflammation, all resulting in syncytiotrophoblast stress, thus leading to maternal (and placental) syndrome [13,48,49,50,51].

Preeclampsia has been proposed as a two-step disease [52]. The first step, starting at early gestation, is characterized by an abnormal trophoblastic invasion with an impaired uterine spiral artery remodeling, leading to placental oxidative stress with hypoxic injury. Systemic and excessive inflammatory processes lead to an increased secretion of pro-inflammatory cytokines, and subsequently to an angiogenic imbalance with an increase in anti-angiogenic factors and a decrease in pro-angiogenic factors in the maternal circulation; this first step is asymptomatic. Especially the angiogenic imbalance seems to provoke the multisystem endothelial dysfunction, leading to the specific clinical features, the second and symptomatic step of preeclampsia after 20 weeks of gestation [53].

EOP and LOP do not differ only in onset and type of clinical manifestations, but also—for the most part—in the pathophysiological mechanisms. After the abnormal trophoblastic invasion at early gestation, the “stressed” syncytiotrophoblast releases pro-inflammatory cytokines, anti-angiogenic agents, and cell-free fetal DNA into the maternal circulation, leading to a systemic inflammatory response [54,55]. In EOP, mainly the defective remodeling of the uterine spiral arteries is responsible for the uteroplacental malperfusion [56]. In LOP, the maternal perfusion is normal, but pre-pregnancy constitutional/metabolic factors, especially obesity, arterial hypertension, metabolic syndrome, and maternal age, play a major role in the pathophysiological pathways [57,58].

In conclusion, late onset preeclampsia seems to be caused by an impaired pre-pregnancy cardiovascular or metabolic status [57].

#### 2.1.2. Prevalance of aPLs in Women with Preeclampsia

Recurrent abortions are the most common complications of OAPS; preeclampsia occurs much less. A significant association between aPLs and recurrent abortions as well as late fetal loss and IUFD was detected in various studies [6,7,8,9]. In contrast, there is a huge difference regarding the association of APS and preeclampsia and the association of (only) aPLs and preeclampsia; 20–50% of women with APS will develop preeclampsia, in the majority of cases severe preeclampsia, whereas “only” up to 10% of women with preeclampsia will test positive for aPLs [39,40,41,42,43,59,60].

Additionally, the prevalence of preeclampsia in women with APS differs according to a range of authors [5,61,62,63].

Clark, E.A. et al. aimed to reveal an association of aPLs and preeclampsia in the absence of other clinical criteria of APS [16]. The authors could not find a relationship between preeclampsia and aPLs in the general population as well as in women at risk for preeclampsia; nevertheless, women who meet the (clinical and laboratory) criteria of APS are at a high risk for developing preeclampsia [16]. According to the authors, testing for aPLs in women who developed early onset preeclampsia can be considered; on the other hand, testing in women who suffered from late-onset preeclampsia is not recommended [16].

Other authors revealed different results, thus confirming an increased risk of preeclampsia already in women with aPLs, but without classic APS [64,65,66,67,68,69]. Heilmann, L. et al. analyzed blood samples from women with preeclampsia (EOP and LOP) and compared the results with women after uncomplicated pregnancy [70]. A total of 20% of women with EOP were positive for at least one aPL, whereas aPLs were present in 6% of women with LOP and in 3% of women with uncomplicated pregnancies [70]. According to these results, testing for aPLs can also be recommended in women with preeclampsia after 34 weeks of gestation [70].

Marchetti, T. et al. analyzed plasma samples from 199 women with “non-severe” preeclampsia and from 143 women with severe preeclampsia, as well as from 195 control women [71]. The authors aimed to determine an incidence of aPLs in women with preeclampsia, “non-severe” and severe. Only in women who developed severe preeclampsia was an association with aPLs detected [71].

In conclusion, the prevalence of aPLs in preeclampsia is not well established yet [3].

### 2.2. Clinical Phenotypes of OAPS

The EUROAPS is a European registry aimed at analyzing the clinical features, laboratory data, and pregnancy (maternal and fetal/neonatal) outcome in a cohort of 1000 women with OAPS [5]. The most prevalent obstetric complications were recurrent miscarriages (38.6%), followed by fetal loss (25.3%), prematurity (28.5%), and stillbirth (23%). EOP was seen in 18.1% of all cases; LOP was present in 4.6% [5].

Cervera, R. et al. observed 1000 patients with APS over a 10-year time period [72]. Prematurity occurred in 48.2% of all live births, and FGR was detected in 26.3%. Preeclampsia was present in only 6.4% [72]. Interestingly, the authors noted that the prevalence of preeclampsia decreased from the first 5 years to the last 5 years from 7.6% to 4.8%. It might be speculated that this phenomenon might be due to a better understanding of the disease and especially an improved management during pregnancy over time [72].

Another more recent work from the same authors investigated several clinical and immunological manifestations in a cohort of 1000 patients with APS in order to define a probable pattern of the disease expression [62]. A total of 71.9% of all included women in the study were pregnant at least one time. The most common obstetric complication was early fetal loss (35.4%), followed by late fetal loss (16.9%), Prematurity occurred in 10.6%, preeclampsia in 9.5%, eclampsia in 4.4%, and placental abruption in 2% [62].

#### Spontaneous Preterm Delivery

Interestingly, women with APS also have an increased risk for preterm delivery, even in the absence of FGR and preeclampsia, and therefore are not indicated but spontaneous.

Yang, J. et al. aimed to analyze—apart from obstetric outcome—risk factors that are associated with pregnancy morbidity in women with APS [73]. The authors revealed a significant higher rate of preeclampsia, as well as premature rupture of fetal membranes and postpartum hemorrhage, in women with APS compared with the control group [73].

The NOH-APS (Nimes Obstetricians and Hematologists Antiphospholipid Syndrome) study analyzed the pregnancy outcomes among women with APS but without a history of thrombosis and compared the prevalence of complications during new pregnancies between treated women with APS and women negative for aPLs (controls) [23]. The authors revealed that among women with APS, fetal loss in a prior pregnancy was a risk factor for preeclampsia and premature birth [23].

A similar observation was made by Deguchi, M. et al. [74]. This multicenter study aimed to investigate the clinical features of APS-complicated pregnancies and to reveal an association of certain clinical factors with adverse pregnancy outcome. Besides thrombocytopenia and positive tests for two or more aPLs, recurrent miscarriages in a previous pregnancy were independent risk factors for premature delivery before 34 weeks of gestation [74].

Other authors revealed similar results [5,62,63,72,75,76,77].

A disbalance of angiogenic and anti-angiogenic factors, which is typical for pregnancy complications characterized by placental dysfunction such as preeclampsia, FGR, or even fetal death, might even occur in a subset of women with spontaneous preterm labor and lacking signs of preeclampsia [78]. It seems that angiogenic factors participate in the pathophysiology of preterm labor in some patients.

Based on all these findings, the transvaginal ultrasound measurement of cervical length should therefore be incorporated into the therapeutic concept in women with APS, especially with prior fetal loss, even when they are asymptomatic (without spontaneous regularly occurring preterm labor).

In singleton pregnancies, the median cervical length measured before 22 weeks of gestation using transvaginal ultrasound is >40 mm; between 22 and 32 weeks of gestation, the median cervical length is approximately 40 mm; and after 32 weeks of gestation, it is about 35 mm [10,79].

A cervical length < 25 mm is considered shortened if the measurement (with transvaginal ultrasound) is performed before 34 weeks of gestation [80]. Primary prevention in the subsequent pregnancy consists of the administration of progesterone from 16 weeks until 36 weeks of gestation (orally 200–400 mg/daily or vaginally 100–200 mg/daily) [81,82,83,84,85], as well as the placement of a cerclage with total closure of the cervix, carried out at the beginning of second trimester (about 15 weeks of gestation) [86,87,88,89].

### 2.3. Other Forms of OAPS

The Sydney criteria of the “classical” OAPS contain at least three consecutive early fetal losses before 10 weeks of gestation, at least one “late” fetal loss after 10 weeks of gestation, and at least one stillbirth or prematurity before 34 weeks of gestation due to preeclampsia, eclampsia, and/or placental insufficiency (fetal growth restriction/FGR) [1].

However, many cases do not fulfill the criteria and are classified as “aPL-related obstetric morbidity” (OMAPS), “non-criteria OAPS” (NC-OAPS), or “sero-negative” APS (SNAPS) [3,4,90,91,92,93,94]. Cases with incomplete clinical or laboratory data are classified as OMAPS or NC-OAPS, whereas SNAPS describes an entity with lacking laboratory criteria but fulfilled clinical criteria. However, other aPL might be present or found in these cases. The most analyzed aPL are those against phosphatidylethanolamine, phosphatidic acid, phosphatidylserine, phosphatidylinositol, vimentin/cardiolipin complex, and annexin A5 [91,92,94,95,96].

OMAPS is defined by (1) (only) two consecutive early fetal losses before 10 weeks of gestation, (2) at least three non-consecutive early fetal losses before 10 weeks of gestation, (3) preeclampsia/eclampsia after 34 weeks of gestation (late onset preeclampsia), (4) placental abruption, (5) preterm delivery after 34 weeks of gestation (late preterm delivery), (6) premature rupture of fetal membranes (PROM), and (7) unexplained recurrent implantation failure in vitro fertilization (failure of at least 3 embryo transfers of good embryo “quality”) [3].

### 2.4. Sequela of Preeclampsia, i.e., Prematurity

It has been demonstrated that complications at early and late gestation have lifelong health consequences for both mother and child [97].

Preeclampsia is associated with maternal and neonatal adverse outcomes: the (iatrogenic) prematurity as well as FGR might lead to severe complications for the offspring; the mother has an increased risk of cardiovascular morbidity and mortality in later life, which was already described in 1964 by Epstein [98]. According to the literature, 30% of women with preeclampsia develop an arterial hypertension and 25% a metabolic syndrome [99,100]. Structural and functional cardiovascular changes were found in women 1 year after preeclamptic pregnancies [101]. Additionally, preeclampsia is associated with several chronic renal disorders in later life [102].

It has also been suggested that preeclampsia might result in a deterioration to the quality of life and increase the risk of postpartal depression [103,104]. Poel et al. revealed in their work from 2009 that 1/5 of all women with preeclampsia are in need of psychological treatment several years after delivery [105].

FGR is associated with increased perinatal morbidity and mortality rates. Several complications such as fetal death, peripartal asphyxia, meconium aspiration, neonatal hypoglycemia, and hypothermia, as well as abnormal neurological development, have been described by various authors [106,107]. However, an increased risk of cardiovascular disease and the related disorders, hypertension, stroke, and type-2 diabetes, later in life are also correlated with FGR [108,109,110].

Premature birth is defined as delivery at less than 37 weeks of gestation and is a significant cause of infant and child morbidity and mortality, especially when occurring before 28 weeks of gestation [111].

Approximately 65% of preterm births are due to spontaneous preterm labor or preterm rupture of fetal membranes (PROM); the remaining are indicated due to maternal and/or fetal reasons [112].

Approximately two-thirds of perinatal mortality and half of long-term neurologic disabilities, such as cerebral palsy, are associated with preterm delivery [113]. It also accounts for the increasing numbers of intergenerational non-communicable diseases [111].

### 2.5. Management of “Classic APS”

The gold standard treatment is low-dose aspirin (LDA; 100–150 mg/day) and low-molecular-weight heparin (LMWH) in prophylactic dosage, starting as early as possible in pregnancy [2]. Some authors recommend a start with LDA already before conception and with LMWH from the moment of positive pregnancy test [114].

However, approximately 20–30% of women will suffer from complications during pregnancy, in spite of treatment with LDA and LMWH.

An interesting fact is that the treatment is more efficient for the prevention of preeclampsia or other forms of placental dysfunction in comparison to recurrent fetal loss [115,116].

Although EOP can be prevented by the use of LDA in most cases, when started before 16 weeks of gestation, this unfortunately does not apply to LOP. “Life-style modification”, such as optimizing pre-pregnancy weight, might be a promising approach [57].

Regarding OMAPS, NC-OAPS, and SNAPS, there is still debate about the management during pregnancy [3].

### 2.6. Management of Refractory APS

Options for the prevention of recurrent fetal loss in women refractory to LDA and LMWH include low-dose prednisolone [116], as well as the use of intravenous immunoglobulins [25,117,118,119,120,121,122,123,124], hydroxychloroquine/HCQ [25,125,126,127,128,129,130,131,132], plasma exchange [24,25,133,134,135], statins [3,136,137], or even biologics, such as TNF-alpha blockers [138,139].

There is lack of data regarding the prevention and the optimal treatment of women with preeclampsia. Van Hoorn, M.E. et al. aimed to reveal a difference in obstetric outcome in aPL-positive pregnant women with preeclampsia and a previous preterm delivery before 34 weeks of gestation in women with hypertensive disorders and/or small-for-gestational-age birthweight who were treated with LMWH and LDA compared to LDA alone. The use of LMWH showed no benefit [140].

The TIPPS trial, a multicenter trial of 292 women with various thrombophilic diseases, analyzed the efficacy of dalteparin versus no dalteparin in the prevention of several complications during pregnancy, such as preeclampsia, placental abruption, or FGR. The results were unable to reveal a potential effect of the prophylactic use of dalteparin [141].

A meta-analysis aimed to prove an effect of LMWH in pregnancies without thrombophilia in order to prevent preeclampsia and FGR. The use of LMWH was associated with a risk reduction of preeclampsia and FGR according to the results; additionally, the use of dalteparin was is correlated with a risk reduction for preeclampsia and FGR. On the other hand, enoxaparin was only associated with a risk reduction for preeclampsia but not FGR. The authors concluded that LMWH has a modest beneficial effect of LMWH for secondary prevention of preeclampsia and FGR [142].

The potential use of LMWH in pregnant women with APS has been verified by many investigations, and the treatment during pregnancy is recommended by international professional guidelines; however, the same cannot be stated for inherited thrombophilia such as factor V Leiden; prothrombin G20210A mutation; and deficiencies of antithrombin, protein C, or protein S. All of these thrombophilias are also associated with pregnancy loss [143,144,145,146].

The ALIFE2 study aimed to assess the use of LMWH versus standard care in women with recurrent pregnancy loss (>2 consecutive miscarriages, non-consecutive, or IUFD) in women with inherited thrombophilia. Inherited thrombophilia included Factor V Leiden mutation (heterozygous and homozygous), prothrombin gene mutation (G20210A), antithrombin deficiency, protein C deficiency, or protein S deficiency. The live birth rate was 72% in the LMWH group and 71% in the standard care group, and thus no significant differences could be found. The authors concluded that the use of LMWH does not result in a higher live birth rate in women with inherited thrombophilia [147].

#### Hydroxychloroquin

HCQ is wieldy used in the prevention and treatment of different autoimmune diseases, such as systemic lupus erythematosus [148], rheumatoid arthritis [149], and Sjögren’s syndrome [150,151,152]. Several studies [125,153,154,155] have provided evidence confirming the safety of HCQ during pregnancy. The positive effect in treatment of APS and reduction of preeclampsia by HCQ has been shown in a few retrospective studies and in vitro experiments. Three retrospective studies connected HCQ with increased live birth in APS patients [129,155,156]. The first one was conducted by Mekinian et al. and involved 14 patients with refractory obstetrical APS who were additionally taking HCQ. As a result, they observed an increase in live births in 78% of patients [129]. Another retrospective study by Sciascia et al. and Gerde et al. also associated HCQ treatment with a higher rate of live births (67%), as well as with a prolongation of pregnancy duration [155,156]. It has been shown that HCQ lowers the incidence of thrombosis in patients with primary APS (PAPS) [157,158]. Moreover, HCQ use was also linked to reductions in IgG anti-cardiolipin, IgG/IgM anti-β2-glycoprotein [157,158], and type I interferon scores [159] in patients with PAPS.

A meta-analysis found an association between lower risk of preeclampsia and HCQ in SLE patients [160,161] but not in aPL/APS patients [161]. Contrary to that, Latino et al. showed that high-risk APS patients treated with HCQ had a significantly lower rate of severe early onset preeclampsia compared to those treated without HCQ [162]. Moreover, Sciascia found out that APS pregnant patients who received HCQ in addition to standard therapy had a lower prevalence of placenta-mediated complications, including PE [155]. Due to that, four prospective clinical trials focused on APS and HCQ (HYPATIA [132], HYDROSAPL [131], HIBISCUS Belizna [130], and a BBQ study [163]) and two focusing on preeclampsia and HCQ (NCT04755322 and NCT05287321) are currently ongoing. These ongoing studies aim to further evaluate the efficacy and safety of HCQ in the management of APS and preeclampsia.

### 2.7. Supplementation as a Potential Additional Treatment

Several additional attempts have been proposed: folic acid supplementation, calcium, and vitamin D being among them [3,164].

#### 2.7.1. Vitamin D

A role of vitamin D deficiency in the development of autoimmune diseases has been proposed by various authors [165].

According to the literature, vitamin D supplementation is associated with an improvement of inflammatory properties, immunological function, and a reduction in antibodies [166].

This observation is supported by population-based studies [167,168,169]. Increased activity of regulatory T cell (Tregs), a subset of T lymphocytes, and decreased proinflammatory (CD4+IL-17AT) cells are observed during vitamin D supplementation, thus leading to an anti-inflammatory T cell profile [170]. Treg cells are important in repressing cytotoxic T cells, Th1 cells, macrophages, and DC and NK cells, leading to immune quiescence during implantation [171]. Thus, vitamin D acts as a natural immune modulator [172].

The modulation functions of vitamin D on adaptive immune cells by increasing the production of IL-1-beta and IL-8 in macrophages and neutrophils is well known. Interestingly, the phagocytic ability of these cells was found to be suppressed due to vitamin D [173]. This may explain the significant association between vitamin D deficiency and autoimmune diseases, such as systemic lupus erythematosus (SLE) and APS [28,30,165,167,174,175,176,177,178,179].

Although vitamin D deficiency is commonly found in pregnant women [180], adverse pregnancy outcomes such as preeclampsia or FGR are still associated with lower levels of vitamin D during pregnancy [181].

According to the literature, vitamin D deficiency in OAPS is as high as 50% [182]; it is also associated with adverse pregnancy outcomes such as placental dysfunction [183].

Cyprian et al. aimed to describe a correlation between vitamin D levels and fertility and pregnancy outcomes, as well as it being a marker of APS disease activity, namely, the presence of flares and complement C3 consumption. The authors revealed that 80% of patients had low vitamin D levels and that 23% of women with hypovitaminosis developed preeclampsia. There were no cases of preeclampsia in women with normal vitamin D levels. Additionally, vitamin D levels below 30 ng/mL were associated with complement activation and a higher incidence of flares [30].

Whether vitamin D supplementation in OAPS is able to modulate inflammatory and immunological function and thus improve disease severity is still unclear.

But, given the described pathophysiological changes in OAPS and the beneficial effects of vitamin D supplementation in modulating the immune system by preventing inflammation, and therefore contributing to the protection of maternal and fetal health, as well as numerous clinical benefits, vitamin D supplementation in OAPS patients is recommended, even though there are still no controlled randomized trials.

#### 2.7.2. Curcumin

Curcumin (1,7-bis(4-hydroxy-3-methoxyphenyl)-1,6-heptadiene-3,5-dione) is the main natural polyphenol present in turmeric [184]. It has been used for medical reasons for approximately 4000 years due to its antimicrobial [185,186], antioxidant, and anti-inflammatory properties [187,188,189]. Furthermore, it has been proposed that turmeric has several beneficial effects in metabolic syndrome [190], as well as diabetes mellitus, and it seems to have analgetic qualities [191].

Clinical studies have confirmed the safety of curcumin. Thus, doses of up to 8000 mg/day are considered safe; above 10,000 mg, several adverse effects, such as diarrhea and yellow stools, occur intermittently [192,193]. However, doses of up to 12,000 mg/day are not reported as toxic. The FDA (Food and Drug Administration) has approved the three main active ingredients of curcuma, namely, curcumin, demethoxycurcumin, and bisdemethoxycurcumin, available in different dosage forms, as “Generally Recognized As Safe (GRAS)” [194]. According to several in vitro studies, curcumin has neither a negative effect on chromosomes nor does it have a mutagenic activity [195].

The majority of curcumin’s probable effects can be explained by its antioxidant and anti-inflammatory properties [188].

Curcumin enhances the serum activities of antioxidants such as SOD (superoxide dismutase), SOD catalase, and lipid peroxides. Furthermore, curcumin can inhibit various forms of free radicals, including reactive oxygen (ROS) and nitrogen species (RNS). ROS-producing enzymes such as cyclooxygenase, which play an important role in the development of preeclampsia, and lipoxygenase are effectively inhibited by curcumin [196,197].

Oxidative stress is closely associated to that of inflammation through similar pathological processes. Inflammatory cells release reactive species, which in turn lead to oxidative stress. ROS and RNS are capable of triggering intracellular signaling cascades that increase the expression of pro-inflammatory genes [190,198].

Inflammation has been observed in many chronic diseases such as cardiovascular disease, metabolic syndrome, diabetes, obesity, and arthritis [199]. Curcumin lowers the raised levels of ROS, IL-6, and TNF-α, and it reduces COX-2 expression. It also significantly reduces the expression of proinflammatory cytokines [200,201].

It is a known fact that elevated levels of TNF-alpha play a significant role in mediating inflammation in preeclampsia as well as APS [202].

Inflammatory cytokines and various inflammatory stimuli activate both TNF-α and NF-κB, which are TNF-α regulators. Studies have shown that curcumin is able to interact with TNF-α and NF-κB and might therefore have an important role in suppressing nuclear translocation of NF-κB in rat vascular smooth muscle cells [190].

Furthermore, it has been shown that curcumin could significantly attenuate anti-β2GPI-induced tissue factor expression in the aorta and peritoneal macrophages. Anti-β2GPI was inhibited in vivo through the activation of NF-κB signaling pathways. It might be speculated that curcumin minimizes the thrombogenic effects of anti-β2GPI [203].

Curcumin has hardly been tested as a sole substance in mouse models in OAPS, rather, it is tested in combination with vitamin D or fish oil in different concentrations [204].

## 3. Conclusions

Although preeclampsia is not the most prevalent obstetric complication in women with APS, its significance in prevention and management as well as regarding an expectable impact on further life for mother and child should not be underestimated, especially in cases of LOP, as these women have mostly an impaired pre-pregnancy cardiovascular or metabolic status.

According to several findings and theories, two considerations appear:

First, although the pathophysiologic similarities of APS and EOP are quite interesting, it is still obtuse why some women with APS develop early onset preeclampsia and others do not.

Second, LOP in women with APS—according to the pathophysiology—may not be caused by aPLs, but it seems to be random and a “side effect”. Additionally, there is still tenuous information regarding the incidence of LOP in women with APS in the literature.

Spontaneous preterm delivery, mostly caused by a premature rupture of fetal membranes, seems to happen not only in rare cases in women with APS. This complication appears mostly by surprise. Regarding the—unfortunately not that seldom—severe consequences for the child, regular screening for a premature rupture of fetal membranes, as well as preterm delivery such as measurements of cervical length, should therefore be implemented in the management during pregnancy in women with APS.

As prior fetal loss seems to be an important risk factor for (spontaneous) preterm delivery, the knowledge of the medical history of previous pregnancies might improve the pregnancy outcome.

The gold standard treatment is still LDA (100–150 mg/day) and LMWH in prophylactic dosage, starting as early as possible in pregnancy. However, the potential prophylactic use of LDA does only apply for EOP; LOP might be prevented by lifestyle modification prior conception.

Regarding OMAPS, NC-OAPS, and SNAPS, there is still debate about the management during pregnancy.

Several additional treatment options for refractory APS have been proposed. However, to this day, only corticosteroids and HCQ are the most feasible therapy options for recurrent refractory cases, in relation to side effects, costs, and practicality.

Nutritional interventions, such as intake of vitamin D, have shown promising beneficial effects [28,29,30,31,32].

Although there are no real data and evidence for curcumin supplementation in APS, nor in OAPS, certain effects of curcumin should not be underestimated.

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
