# Peer review of "Preeclampsia and the Antiphospholipid Syndrome"

_biomedicines, 2023, doi:10.3390/biomedicines11082298_

Round 1

Reviewer 1 Report

It is a complete review of the current state of the art in OAPS. This manifestation may be included as a more relevant clinical criterion in future classifications. I will highlight the excellent description of PE pathophysiology. Here I recommend some modifications to improve the understanding: 

- A description of the methods and the strategies used to collect the bibliography consulted for the paper. I suppose it wasn't systemically and only follows the author's criteria. 

- The definition of PAPs and SAPs may not be needed, those concepts are not used in the manuscript and the differentiation does not really suggest a different treatment for the APS.

- In line 88 I don't understand why the authors defined placental dysfunction as iatrogenic. 

- Line 99: The manuscript will benefit from a description of the consequences after delivery, future readers of areas different from the obstetrics may need it.

- The sentence in line 157 may be confusing, the authors mentioned the PE prevalence in APS but in the previous paragraph they establish this as 20-50%. Maybe the second paragraph is related to the prevalence un aPL positive patients (this is also relevant).

- The second paragraph of section 2.3 repeat a information previously mentioned.

- Is relevant to mention the aPL related to OAPS but out of the classification criteria (anti-fosfatidilserne prothrombin, annexin... )

- Line 247: SNAPS is related to lacking laboratory criteria but with clinical criteria suggestive of APS (Pires da Rosa G, Bettencourt P, Rodríguez-Pintó I, Cervera R, Espinosa G. "Non-criteria" antiphospholipid syndrome: A nomenclature proposal. Autoimmun Rev. 2020 Dec;19(12):102689. doi: 10.1016/j.autrev.2020.102689. Epub 2020 Oct 22. PMID: 33223008.)

- Section 2.7.1 paragraph 4: The mentioned cells are part of the innate immune system. The IL produce have a more defined inflammatory effect. Mention the adaptative immune system is not accurate. 

- Any studies with curcumin in APS?

- First paragraph of the discussion: In my understanding, the impact of APS may be especially in EOP, not in LOP. 

Minor suggestions:

- Line 118: Doble space after the point. 

- The sentences between line 184 and line 190 are describing the same paper of Cercera et al. The unification of these sentences in the same paragraph may be useful for readers. 

- Subtitle 2.2.1 Change "spontaneous" for "Spontaneous".

- There is any reason for the absence of words in line 232?

- Line 260: long space between the words "fetal" and "growth".

- Line 308: Separate PE-de Von Hoorn

- Line 383: Long space after (141)

Minor suggestions are recommended in the previous section. 

Author Response

Thank you for your helpful and valuable comments and suggestions!

Reviewer 1:

  • The definition of PAPs and SAPs may not be needed, those concepts are not used in the manuscript and the differentiation does not really suggest a different treatment for the APS

Has been deleted in the text.

  • In line 88 I don't understand why the authors defined placental dysfunction as iatrogenic. 

Not the placental dysfunction is iatrogenic, but preterm delivery/prematurity is.

The only causative treatment for preeclampsia is delivery, irrespective of the gestational age. Hence preterm delivery in preeclampsia is not due to premature rupture of fetal membranes or preterm labour.

  • Line 99: The manuscript will benefit from a description of the consequences after delivery, future readers of areas different from the obstetrics may need it.

These consequences are described under “2.4. Sequela of preeclampsia, respectively prematurity”- line 262 to line 289

  • The sentence in line 157 may be confusing, the authors mentioned the PE prevalence in APS but in the previous paragraph they establish this as 20-50%. Maybe the second paragraph is related to the prevalence un aPL positive patients (this is also relevant). Exactly! We wanted to emphasize the difference regarding the prevalence of PE in women with APS and the frequency of aPl in women with PE. Up to 20-50% of women with APS might develop PE, in contrast only 10% of women who developed PE during pregnancy will test positive for at least one aPl.
  • It is relevant to mention the aPL related to OAPS but out of the classification criteria (anti-fosfatidilserne prothrombin, annexin...). Has been modified in the manuscript.
  • Line 247: SNAPS is related to lacking laboratory criteria but with clinical criteria suggestive of APS (Pires da Rosa G, Bettencourt P, Rodríguez-Pintó I, Cervera R, Espinosa G. "Non-criteria" antiphospholipid syndrome: A nomenclature proposal. Autoimmun Rev. 2020 Dec;19(12):102689. doi: 10.1016/j.autrev.2020.102689. Epub 2020 Oct 22. PMID: 33223008.). The article has been cited in the manuscript.
  • Section 2.7.1 paragraph 4: The mentioned cells are part of the innate immune system. The IL produce have a more defined inflammatory effect. Mention the adaptative immune system is not accurate. 
  • Any studies with curcumin in APS? Unfortunately, no
  • First paragraph of the discussion: In my understanding, the impact of APS may be especially in EOP, not in LOP. I think so as well, but LOP might even happen in APS- according to some authors. And the consequences of LOP should not be underestimated.
  • Line 118: Doble space after the point.
  • The sentences between line 184 and line 190 are describing the same paper of Cercera et al. The unification of these sentences in the same paragraph may be useful for readers.
  • Subtitle 2.2.1 Change "spontaneous" for "Spontaneous". Done
  • There is any reason for the absence of words in line 232?
  • Line 260: long space between the words "fetal" and "growth". Modified.
  • - Line 308: Separate PE-de Von Hoorn.
  • - Line 383: Long space after (141).

Reviewer 2 Report

1.  These authors have written a review article on the association between preeclampsia and the antiphospholipid syndrome.  I have reviewed this article from the perspective of an internist who works in the medical intensive care unit.  I do not have a background in either obstetrics or hematology.
2.  At the end of the introduction, the authors suggest that they want to answer 3 questions about the relationship between preeclampsia and antiphospholipid syndrome.  I would encourage them to make these answers clear in the discussion.  The only question I am certain about is whether or not there are additional treatment options.  In the abstract and in the first paragraph of the discussion, they state the reasons for prematurity, respectively preterm delivery are mostly iatrogenic due to placental dysfunction, such as PE or FGR.  It is not clear to me why these would be considered iatrogenic reasons for prematurity.

3. The authors use many abbreviations in their text.  For most clinicians these abbreviations are unfamiliar.  They should try to limit those the use of those abbreviations.  In addition, they frequently give the abbreviations such as low molecular weight heparin (LMWH) and then periodically used the same phrase again in the text rather than just LMWH.  If they plan to use abbreviations, they should consistently use them throughout the text.
4.  The authors use some words and phrases that seem unusual for a medical manuscript.  These would include the word nay, the words in a nutshell, diverse authors, bona fide, and the abbreviation i.a.
4.  In the paragraph entitled Thinking outside the box, it is not clear what thinking in this section

represents an unusual approach or understanding of the information being discussed.
5.  In the section called Management of “classical APS”, the authors include 2 paragraphs which are highlighted in yellow which discuss the management of patients with other coagulopathies.  This information is not particularly relevant to their main discussion.
6.  In the paragraph on curcumin, they suggest the doses at 8.000 mg/day are considered safe.  Usually, that number would be written 8,000, at least in United States.
7.  It might be helpful if the authors included some box diagrams to illustrate the relationships between APS and obstetrical outcomes and possibly between antiphospholipid antibodies and obstetrical outcomes.

Limit the use of some unconventional nonmedical language in the manuscript.  Make the use of abbreviations and uniform.

Author Response

Reviewer 2:

Thank you for your helpful and valuable comments and suggestions!

  1. The following questions were answered in the text.
  • Especially, does prematurity occur without PE or other forms of placental dysfunction? Under2.1. Spontaneous preterm delivery and in the conclusion
  • Does late onset preeclampsia in women with APS (or aPl positive women) even exist? If so, should the Sydney criteria be adapted? : “Second, LOP in women with APS- according to the pathophysiology- may not be caused by aPl, but seems to be random and a “side effect”. Additionally, there is still tenous information regarding the incidence of LOP in women with APS in literature”

As there are some considerations in changing the Sydney criteria regarding late onset PE, we aimed to review how often late onset PE really occurs according to literature. We were surprised about the lack of information. Only the EUROAPS group described some cases of late onset PE. We described this problem in the “conclusion” section: Second, LOP in women with APS- according to the pathophysiology- may not be caused by aPl, but seems to be random and a “side effect”. Additionally, there is still tenous information regarding the incidence of LOP in women with APS in literature.

But we have to admit that some of the questions were not answered adequately, therefore we modified the text at the end of the introduction.

  1. In the abstract and in the first paragraph of the discussion, they state the reasons for prematurity, respectively preterm delivery are mostly iatrogenic due to placental dysfunction, such as PE or FGR.  It is not clear to me why these would be considered iatrogenic reasons for prematurity. Not the placental dysfunction is iatrogenic, but preterm delivery/prematurity is.

The only causative treatment for preeclampsia is delivery, irrespective of the gestational age. Hence preterm delivery in preeclampsia is not due to premature rupture of fetal membranes or preterm labour.

  1. The authors use many abbreviations in their text.  For most clinicians these abbreviations are unfamiliar.  They should try to limit those the use of those abbreviations.  In addition, they frequently give the abbreviations such as low molecular weight heparin (LMWH) and then periodically used the same phrase again in the text rather than just LMWH.  If they plan to use abbreviations, they should consistently use them throughout the text. Has been modified in the text.
  2. The authors use some words and phrases that seem unusual for a medical manuscript.  These would include the word nay, the words in a nutshell, diverse authors, bona fide, and the abbreviation a. Has been modified in the text.
  3. In the paragraph entitled Thinking outside the box, it is not clear what thinking in this section represents an unusual approach or understanding of the information being discussed. The title has been changed.
  4. In the section called Management of “classical APS”, the authors include 2 paragraphs which are highlighted in yellow which discuss the management of patients with other coagulopathies.  This information is not particularly relevant to their main discussion. We were asked by the editor to include this reference.
  5. In the paragraph on curcumin, they suggest the doses at 8.000 mg/day are considered safe.  Usually, that number would be written 8,000, at least in United States. Has been modified in the text.

Round 2

Reviewer 2 Report

Thank you for your changes and revision.I think the current version improves the presentation of the information.

It will need routine copy editing